# ContraGAN: Contrastive Learning for Conditional Image Generation

**Minguk Kang**        **Jaesik Park**

Graduate School of Artificial Intelligence
POSTECH
{mgkang, jaesik.park}@postech.ac.kr

## Abstract

Conditional image generation is the task of generating diverse images using class label information. Although many conditional Generative Adversarial Networks (GAN) have shown realistic results, such methods consider pairwise relations between the embedding of an image and the embedding of the corresponding label (*data-to-class relations*) as the conditioning losses. In this paper, we propose ContraGAN that considers relations between multiple image embeddings in the same batch (*data-to-data relations*) as well as the data-to-class relations by using a conditional contrastive loss. The discriminator of ContraGAN discriminates the authenticity of given samples and minimizes a contrastive objective to learn the relations between training images. Simultaneously, the generator tries to generate realistic images that deceive the authenticity and have a low contrastive loss. The experimental results show that ContraGAN outperforms state-of-the-art-models by 7.3% and 7.7% on Tiny ImageNet and ImageNet datasets, respectively. Besides, we experimentally demonstrate that contrastive learning helps to relieve the overfitting of the discriminator. For a fair comparison, we re-implement twelve state-of-the-art GANs using the PyTorch library. The software package is available at https://github.com/POSTECH-CVLab/PyTorch-StudioGAN.

## 1   Introduction

Generative Adversarial Networks (GAN) [1] have introduced a new paradigm for realistic data generation. Many approaches have shown impressive improvements in un/conditional image generation tasks [2, 3, 4, 5, 6, 7, 8, 9]. The studies on non-convexity of objective landscapes [10, 11, 12] and gradient vanishing problems [3, 11, 13, 14] emphasize the instability of the adversarial dynamics. Therefore, many approaches have tried to stabilize the training procedure by adopting well-behaved objectives [3, 13, 15] and regularization techniques [4, 7, 16]. In particular, spectral normalization [4] with a projection discriminator [17] made the first success in generating images of ImageNet dataset [18]. SAGAN [5] shows using spectral normalization on both the generator and discriminator can alleviate training instability of GANs. BigGAN [6] dramatically advances the quality of generated images by scaling up the number of network parameters and batch size.

On this journey, conditioning class information for the generator and discriminator turns out to be the secret behind realistic image generation [17, 19, 20]. ACGAN [19] validates this direction by training a softmax classifier along with the discriminator. ProjGAN [17] utilizes a projection discriminator with probabilistic model assumptions. Especially, ProjGAN shows surprising image synthesis results and becomes the basic model adopted by SNGAN [4], SAGAN [7], BigGAN [6], CRGAN [7], and LOGAN [9]. However, GANs with the projection discriminator have overfitting issues, which lead to the collapse of adversarial training [21, 9, 22, 23]. The ACGAN is known to be unstable when the number of classes increases [17, 19].

In this paper, we propose a new conditional generative adversarial network framework, namely *Contrastive Generative Adversarial Networks* (ContraGAN). Our approach is motivated by an interpretation that ACGAN and ProjGAN utilize *data-to-class* relation as the conditioning losses. Such losses only consider relations between the embedding of an image and the embedding of the corresponding label. In contrast, ContraGAN is based on a conditional contrastive loss (2C loss) to consider *data-to-data* relations in the same batch. ContraGAN pulls the multiple image embeddings closer to each other when the class labels are the same, but it pushes far away otherwise. In this manner, the discriminator can capture not only *data-to-class* but also *data-to-data* relations between samples.

We perform image generation experiments on CIFAR10 [24], Tiny ImageNet [25], and ImageNet [18] datasets using various backbone architectures, such as DCGAN [2], ResGAN [26, 16], and Big-GAN [6] equipped with spectral normalization [4]. Through exhaustive experiments, we verify that the proposed ContraGAN improves the state-of-the-art-models by 7.3% and 7.7% on Tiny ImageNet and ImageNet datasets respectively, in terms of Frechet Inception Distance (FID) [27]. Also, ContraGAN gives comparable results (1.3% lower FID) on CIFAR10 with the art model [6]. Since ContraGAN can learn plentiful data-to-data relations from a properly sized batch, it reduces FID significantly *without hard negative and positive mining*. Furthermore, we experimentally show that 2C loss alleviates the overfitting problem of the discriminator. In the ablation study, we demonstrate that ContraGAN can benefit from consistency regularization [7] that uses data augmentations.

In summary, the contributions of our work are as follows:

- We propose novel Contrastive Generative Adversarial Networks (ContraGAN) for conditional image generation. ContraGAN is based on a novel conditional contrastive loss (2C loss) that can learn both data-to-class and data-to-data relations.

- We experimentally demonstrate that ContraGAN improves state-of-the-art-results by 7.3% and 7.7% on Tiny ImageNet and ImageNet datasets, respectively. Contrastive learning also helps to relieve the overfitting problem of the discriminator.

- ContraGAN shows favorable results without data augmentations for consistency regularization. If consistency regularization is applied, ContraGAN can give superior image generation results.

- We provide implementations of twelve state-of-the-art GANs for a fair comparison. Our implementation of the prior arts for CIFAR10 dataset achieves even better performances than FID scores reported in the original papers.

## 2   Background

### 2.1   Generative Adversarial Networks

Generative adversarial networks (GAN) [1] are implicit generative models that use a generator and a discriminator to synthesize realistic images. While the discriminator ($D$) should distinguish whether the given images are synthesized or not, the generator ($G$) tries to fool the discriminator by generating realistic images from noise vectors. The objective of the adversarial training is as follows:

$$\min_{G} \max_{D} \mathbb{E}_{\mathbf{x} \sim p_{\text{real}}(\mathbf{x})}[\log(D(\mathbf{x}))] + \mathbb{E}_{\mathbf{z} \sim p(\mathbf{z})}[\log(1 - D(G(\mathbf{z})))], \qquad (1)$$

where $p_{\text{real}}(\mathbf{x})$ is the real data distribution, and $p_{\mathbf{z}}(\mathbf{z})$ is a predefined prior distribution, typically multivariate Gaussian. Since the dynamics between the generator and discriminator is unstable, and it is hard to achieve the Nash equilibrium [28], there are many objective functions [3, 13, 15, 29] and regularization techniques [4, 7, 16, 21] to help networks to converge to a proper equilibrium.

### 2.2   Conditional GANs

One of the widely used strategies to synthesize realistic images is utilizing class label information. Early approaches in this category are conditional variational auto-encoder (CVAE) [30] and conditional generative adversarial networks [31]. These approaches concatenate a latent vector with the label to manipulate the semantic characteristics of the generated image. Since DCGAN [2] demonstrated high-resolution image generation, GANs utilizing class label information has shown advanced performances [6, 7, 9, 8].

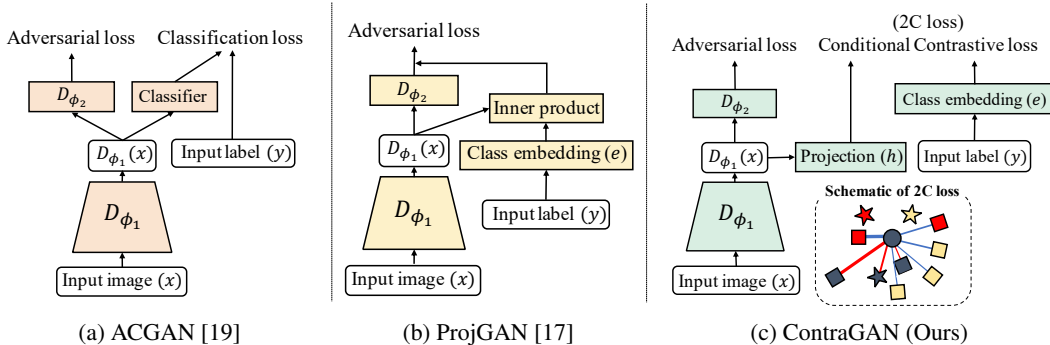

Figure 1: Schematics of discriminators of three conditional GANs. (a) ACGAN [19] has an auxiliary classifier to guide the generator to synthesize well-classifiable images. (b) ProjGAN [17] improves ACGAN by adding the inner product of an embedded image and the corresponding class embedding. (c) Our approach extends ACGAN and ProjGAN with a conditional contrastive loss (2C loss). ContraGAN considers multiple positive and negative pairs in the same batch. ContraGAN utilizes 2C loss to update the generator as well.

The most common approach of conditional GANs is to inject label information into the generator and discriminator. ACGAN [19] attaches an auxiliary classifier on the top of convolutional layers in the discriminator to distinguish the classes of images. An illustration of ACGAN is shown in Fig. 1a. ProjGAN [17] points out that ACGAN is likely to generate easily classifiable images, and the generated images are not diverse. ProjGAN proposes a projection discriminator to relieve the issues (see Fig. 1b). However, these approaches do not explicitly consider data-to-data relations in the training phase. Besides, the recent study by Wu *et al.* [9] discovers that BigGAN with the projection discriminator [6] still suffers from the discriminator's overfitting and training collapse problems.

## 3 Method

We begin with analyzing that conditioning functions of ACGAN and ProjGAN can be interpreted as pair-based losses that look at only data-to-class relations of training examples (Sec. 3.1). Then, in order to consider both data-to-data and data-to-class relations, we devise a new conditional contrastive loss (2C loss) (Sec. 3.2). Finally, we propose Contrastive Generative Adversarial Networks (ContraGAN) for conditional image generation (Sec. 3.3).

### 3.1 Conditional GANs and Data-to-Class Relations

The goal of the discriminator in ACGAN is to classify the class of a given image and the sample's authenticity. Using data-to-class relations, i.e., information about which class a given data belongs to, the generator tries to generate fake images that can deceive the authenticity and are classified as the target labels. Since ACGAN uses a cross-entropy loss to classify the class of an image, we can regard the conditioning loss of ACGAN as a pair-based loss that can consider only data-to-class relations (see Fig. 2d). ProjGAN tries to maximize inner-product values between embeddings of real images and the corresponding target embeddings while minimizing the inner-product values when the images are fake. Since the discriminator of ProjGAN pushes and pulls the embeddings of images according to the authenticity and class information, we can think of the conditioning objective of ProjGAN as a pair-based loss that considers data-to-class relations (see Fig. 2e). Unlike ACGAN, which looks at relations between a fixed one-hot vector and a sample, ProjGAN can consider more flexible relations using a learnable class embedding, namely Proxy.

### 3.2 Conditional Contrastive Loss

To exploit data-to-data relations, we can adopt loss functions used in self-supervised [34] learning or metric learning [32, 35, 36, 37, 38, 39]. In other words, our approach is to *add a metric learning or self-supervised learning objective* in the *discriminator* and *generator* to explicitly control distances between embedded image features depending on the labels. Several metric learning losses, such

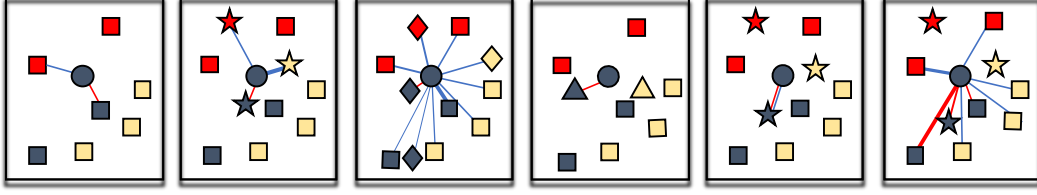

(a) Triplet [32]  (b) P-NCA [33]  (c) NT-Xent [34] (d) ACGAN [19] (e) ProjGAN [17] (f) 2C loss (Ours)

Figure 2: Illustrative figures visualize metric learning losses (a,b,c) and conditional GANs (d,e,f). The color indicates the class label and the shape represents the role. (Square) the embedding of an image. (Diamond) the embedding of an augmented image. (Circle) a reference image embedding. Each loss is applied to the reference. (Star) the embedding of a class label. (Triangle) the one-hot encoding of a class label. The thicknesses of red and blue lines represent the strength of pull and push force, respectively. The loss function of ProjGAN lets the reference and the corresponding class embedding be close to each other when the reference is real, but it pushes far away otherwise. Compared to ACGAN and ProjGAN, 2C loss can consider both data-to-class and data-to-data relations between training examples.

as contrastive loss [35], triplet loss [32], quadruplet loss [36], and N-pair loss [37] could be good candidates. However, it is known that 1) mining informative triplets or quadruplets requires higher training complexity, and 2) poor tuples make the training time longer. While the proxy-based losses [33, 38, 39] relieves mining complexity using trainable class embedding vectors, such losses do not explicitly take data-to-data relations [40] into account.

Before introducing the proposed 2C loss, we bring NT-Xent loss [34] to express our idea better. Let $\boldsymbol{X} = \{\boldsymbol{x}_1, ..., \boldsymbol{x}_m\}$, where $\boldsymbol{x} \in \mathbb{R}^{W \times H}$ be a randomly sampled minibatch of training images and $\boldsymbol{y} = \{y_1, ..., y_m\}$, where $y \in \mathbb{R}$ be the collection of corresponding class labels. Then, we define a deep neural network encoder $S(\boldsymbol{x}) \in \mathbb{R}^k$ and a projection layer that embeds onto a new unit hypersphere $h : \mathbb{R}^k \longrightarrow \mathbb{S}^d$. Then, we can map the data space to the hypersphere using the composition of $l = h(S(\cdot))$. NT-Xent loss conducts random data augmentations $T$ on the training data $\boldsymbol{X}$, and we denote it as $\boldsymbol{A} = \{\boldsymbol{x}_1, T(\boldsymbol{x}_1), ..., \boldsymbol{x}_m, T(\boldsymbol{x}_m)\} = \{\boldsymbol{a}_1, ..., \boldsymbol{a}_{2m}\}$. Using the above, we can formulate NT-Xent loss as follows:

$$\ell(\boldsymbol{a}_i, \boldsymbol{a}_j; t) = -\log\left(\frac{\exp(l(\boldsymbol{a}_i)^\top l(\boldsymbol{a}_j)/t)}{\sum_{k=1}^{2m} \mathbf{1}_{k \neq i} \cdot \exp(l(\boldsymbol{a}_i)^\top l(\boldsymbol{a}_k)/t)}\right), \tag{6}$$

where the scalar value $t$ is a temperature to control push and pull force. In this work, we use the part of the discriminator network $(D_{\phi_1})$ before the fully connected layer as the encoder network $(S)$ and use multi-layer perceptrons parameterized by $\varphi$ as the projection head $(h)$. As a result, we can map the data space to the unit hypersphere using $l = h(D_{\phi_1}(\cdot))$.

However, Eq. (6) requires proper data augmentations and can not consider data-to-class relations of training examples. To resolve these issues, we propose to use the *embeddings of class labels* instead of using data augmentations. With a class embedding function $e(y) : \mathbb{R} \longrightarrow \mathbb{R}^d$, Eq. (6) can be formulated as follows:

$$\ell(\boldsymbol{x}_i, y_i; t) = -\log\left(\frac{\exp(l(\boldsymbol{x}_i)^\top e(y_i)/t)}{\exp(l(\boldsymbol{x}_i)^\top e(y_i)/t) + \sum_{k=1}^{m} \mathbf{1}_{k \neq i} \cdot \exp(l(\boldsymbol{x}_i)^\top l(\boldsymbol{x}_k)/t)}\right). \tag{7}$$

Eq. (7) pulls a reference sample $\boldsymbol{x}_i$ nearer to the class embedding $e(y_i)$ and pushes the others away. This scheme may push negative samples which have the same label as $y_i$. Therefore, we make an exception by adding cosine similarities of such negative samples in the numerator of Eq. (7). The final loss function is as follows:

$$\ell_{2C}(\boldsymbol{x}_i, y_i; t) = -\log\left(\frac{\exp(l(\boldsymbol{x}_i)^\top e(y_i)/t) + \sum_{k=1}^{m} \mathbf{1}_{y_k = y_i} \cdot \exp(l(\boldsymbol{x}_i)^\top l(\boldsymbol{x}_k)/t)}{\exp(l(\boldsymbol{x}_i)^\top e(y_i)/t) + \sum_{k=1}^{m} \mathbf{1}_{k \neq i} \cdot \exp(l(\boldsymbol{x}_i)^\top l(\boldsymbol{x}_k)/t)}\right). \tag{8}$$

Eq. (8) is the proposed conditional contrastive loss (2C loss). Minimizing 2C loss will reduce distances between the embeddings of images with the same labels while maximizing the others. 2C loss explicitly considers the data-to-data relations $l(\boldsymbol{x}_i)^\top l(\boldsymbol{x}_k)$ and data-to-class relations $l(\boldsymbol{x}_i)^\top e(y_i)$ without comprehensive mining of the training dataset and augmentations.

---

**Algorithm 1** : Training ContraGAN

---

**Input:** Learning rate: $\alpha_1, \alpha_2$. Adam hyperparameters [41]: $\beta_1, \beta_2$. Batch size: $m$. Temperature: $t$.
   # of discriminator iterations per single generator iteration: $n_{dis}$. Contrastive coefficient: $\lambda$.
   Parameters of the generator, the discriminator, and the projection layer: $(\theta, \phi, \varphi)$.
**Output:** Optimized $(\theta, \phi, \varphi)$.

---

1: Initialize $(\theta, \phi, \varphi)$
2: **for** $\{1, ..., \text{\# of training iterations}\}$ **do**
3:    **for** $\{1, ..., n_{\text{dis}}\}$ **do**
4:       Sample $\{(\boldsymbol{x}_i, y_i^{\text{real}})\}_{i=1}^m \sim p_{\text{real}}(\mathbf{x}, \mathbf{y})$
5:       Sample $\{\boldsymbol{z}_i\}_{i=1}^m \sim p(\mathbf{z})$ and $\{y_i^{\text{fake}}\}_{i=1}^m \sim p(\mathbf{y})$
6:       $\mathcal{L}_C^{\text{real}} \longleftarrow \frac{1}{m} \sum_{i=1}^m \ell_{2\text{C}}(\boldsymbol{x}_i, y_i^{\text{real}}; t)$                    ▷ Eq. (8) with real images.
7:       $\mathcal{L}_D \longleftarrow \frac{1}{m} \sum_{i=1}^m \{D_\phi(G_\theta(\boldsymbol{z}_i, y_i^{\text{fake}}), y_i^{\text{fake}}) - D_\phi(\boldsymbol{x}_i, y_i^{\text{real}})\} + \lambda \mathcal{L}_C^{\text{real}}$
8:       $\phi, \varphi \longleftarrow \text{Adam}(\mathcal{L}_D, \alpha_1, \beta_1, \beta_2)$
9:    **end for**
10:    Sample $\{\boldsymbol{z}_i\}_{i=1}^m \sim p(\mathbf{z})$ and $\{y_i^{\text{fake}}\}_{i=1}^m \sim p(\mathbf{y})$
11:    $\mathcal{L}_C^{\text{fake}} \longleftarrow \frac{1}{m} \sum_{i=1}^m \ell_{2\text{C}}(G_\theta(\boldsymbol{z}_i, y_i^{\text{fake}}), y_i^{\text{fake}}; t)$         ▷ Eq. (8) with fake images.
12:    $\mathcal{L}_G \longleftarrow -\frac{1}{m} \sum_{i=1}^m \{D_\phi(G_\theta(\boldsymbol{z}_i, y_i^{\text{fake}}), y_i^{\text{fake}})\} + \lambda \mathcal{L}_C^{\text{fake}}$
13:    $\theta \longleftarrow \text{Adam}(\mathcal{L}_G, \alpha_2, \beta_1, \beta_2)$
14: **end for**

---

## 3.3 Contrastive Generative Adversarial Networks

With proposed 2C loss, we describe the framework, called ContraGAN and introduce training procedures. Like the typical training procedures of GANs, ContraGAN has a discriminator training step and a generator training step that compute an adversarial loss. With this foundation, ContraGAN additionally calculates 2C loss using a set of real or fake images. Algorithm 1 shows the training procedures of the proposed ContraGAN. A notable aspect is that 2C loss is computed using $m$ real images in the discriminator training step and $m$ generated images in the generator training step.

In this manner, the discriminator updates itself by minimizing the distances between real image embeddings from the same class while maximizing it otherwise. By forcing the embeddings to relate via 2C loss, the discriminator can learn the fine-grained representations of real images. Similarly, the generator exploits the knowledge of the discriminator, such as intra-class characteristics and higher-order representations of the real images, to generate more realistic images.

## 3.4 Differences between 2C Loss and NT-Xent Loss

NT-Xent loss [34] is intended for unsupervised learning. It regards the augmented image as the positive sample to consider data-to-data relations between an original image and the augmented image. On the other hand, 2C loss utilizes weak supervision of label information. Therefore, compared with 2C loss, NT-Xent hardly gathers image embeddings of the same class, since there is no supervision from the label information. Besides, NT-Xent loss requires extra data augmentations and additional forward and backward propagations, which induce a few times of longer training time than the model with 2C loss.

# 4 Experiments

## 4.1 Datasets and Evaluation Metric

We perform conditional image generation experiments with CIFAR10 [24], Tiny ImageNet [25], and ImageNet [18] datasets to compare the proposed approach with other approaches.

**CIFAR10** [24] is a widely used benchmark dataset in many image generation works [4, 6, 7, 8, 9, 17, 19], and it contains $32 \times 32$ pixels of color images for 10 different classes. The dataset consists of 60,000 images in total. It is divided into 50,000 images for training and 10,000 images for testing.

**Tiny ImageNet** [25] provides 120,000 color images in total. Image size is $64 \times 64$ pixels, and the dataset consists of 200 categories. Each category has 600 images divided into 500, 50, and 50 samples for training, validation, and testing, respectively. Tiny ImageNet has $10\times$ smaller number of images per class than CIFAR10, but it provides $20\times$ larger number of classes than CIFAR10. Compared to CIFAR10, Tiny ImageNet is selected to test a more challenging scenario – the number of images per class is not plentiful, but the network needs to learn more categories.

**ImageNet** [18] provides 1,281,167 and 50,000 color images for training and validation respectively, and the dataset consists of 1,000 categories. We crop each image using a square box whose length is the same as the shorter side of the image. The cropped images are rescaled to $128 \times 128$ pixels.

**Frechet Inception Distance (FID)** is an evaluation metric used in all experiments in this paper. The FID proposed by Heusel *et al.* [42] calculates Wasserstein-2 distance [43] between the features obtained from real images and generated images using Inception-V3 network [44]. Since FID is the distance between two distributions, *lower* FID indicates *better* results.

## 4.2 Software

There are various approaches that report strong FID scores, but it is not easy to reproduce the results because detailed specifications for training or ways to measure the results are not clearly stated. For instance, FID could be different depending on the choice of the reference images (training, validation, or testing datasets could be used). Besides, FID score of prior work is not consistent, depending on TensorFlow versions [45]. Therefore, we re-implement twelve state-of-the-art GANs [2, 13, 15, 3, 16, 10, 19, 17, 4, 5, 6, 7] to validate the proposed ContraGAN under the same condition. Our implementation carefully follows the principal concepts and the available specifications in the prior work. Experimental results show that the results from our implementation are superior to the numbers in the original papers [4, 6] for the experiments using CIFAR10 dataset. We hope that our implementation would relieve efforts to compare various GAN pipelines.

## 4.3 Experimental Setup

To conduct a reliable assessment, all experiments that use CIFAR10 and Tiny ImageNet datasets are performed three times with different random seeds, and we report means and standard deviations of FIDs. Experiments using ImageNet are executed once, and we report the best performance during the training. We calculate FID using CIFAR10's test images and the same amount of generated images. For the experiments using Tiny ImageNet and ImageNet, we use the validation set with the same amount of generated images. All FID values reported in our paper are calculated using the PyTorch FID implementation [46].

Since spectral normalization [4] has become an essential element in modern GAN training, we use hinge loss [15] and apply spectral normalization on all architectures used in our experiments. We adopt modern architectures used in the papers: DCGAN [2, 4], ResGAN [26, 16], and BigGAN [6], and all details about the architectures are described in the supplement.

Since the conditioning strategy used in the generator of ACGAN differs from that of ProjGAN, we incorporate the generator's conditioning method in all experiments for a fair comparison. We use the conditional coloring transform [47, 48, 17], which is the method adopted by the original ProjGAN.

Before conducting the main experiments, we investigate performance changes according to the type of projection layer $h$ in Eq. (8) and batch size. Although Chen *et al.* [34] reports that contrastive learning can benefit from a higher-dimensional projection and a larger batch size, we found that the linear projection with batch size 64 for CIFAR10 and 1,024 for Tiny ImageNet performs the best. For the dimension of the projection layer, we select 512 for CIFAR10, 768 for Tiny ImageNet, and 1,024 for ImageNet experiments. We do a grid search to find a proper temperature ($t$) used in Eq. 8 and experimentally found that the temperature of $1.0$ gives the best results. Detailed hyperparameter settings used in our experiments are described in the supplement.

## 4.4 Evaluation Results

**Effectiveness of 2C loss.** We compare 2C loss with P-NCA loss [33], NT-Xent loss [34], and the objective function formulated in Eq. 7. P-NCA loss [24] does not explicitly look at data-to-data relations, and NT-Xent loss [25] (equivalent to Eq. 6) does not take data-to-class relations into account.

Table 1: Experiments on the effectiveness of 2C loss. Considering both data-to-data and data-to-class relations largely improves image generation results based on FID values. Mean±variance of FID is reported, and lower is better.

| Dataset | Uncond. GAN [6] | with P-NCA loss [33] | with NT-Xent loss [34] | with Eq.7 loss | with 2C loss (ContraGAN) |
|---|---|---|---|---|---|
| CIFAR10 [24] | 15.550±1.955 | 15.350±0.017 | 14.832±0.695 | 10.886±0.072 | **10.597±0.273** |
| Tiny ImageNet [25] | 56.297±1.499 | 47.867±1.813 | 54.394±9.982 | 33.488±1.006 | **32.720±1.084** |

Table 2: Experiments using CIFAR10 and Tiny ImageNet datasets. Using three backbone architectures (DCGAN, ResGAN, and BigGAN), we test three approaches using different class conditioning models (ACGAN, ProjGAN, and ContraGAN (ours)).

| Dataset | Backbone | Method for class information conditioning | | |
|---|---|---|---|---|
| | | ACGAN [19] | ProjGAN [17] | ContraGAN (Ours) |
| CIFAR10 [24] | DCGAN [2, 4] | 21.439±0.581 | 19.524± 0.249 | **18.788±0.571** |
| | ResGAN [26, 16] | 11.588±0.093 | **11.025± 0.177** | 11.334±0.126 |
| | BigGAN [6] | 10.697±0.129 | 10.739±0.016 | **10.597±0.273** |
| Tiny ImageNet [25] | BigGAN [6] | 88.628±5.523 | 37.563±4.390 | **32.720±1.084** |

Table 3: Comparison with state-of-the-art GAN models. We mark '*' to FID values reported in the original papers [4, 5, 7]. The other FID values are obtained from our implementation. We conduct ImageNet [18] experiments with a batch size of 256.

| Dataset | SNResGAN [4] | SAGAN [5] | BigGAN [6] | ContraGAN (Ours) | Improvement |
|---|---|---|---|---|---|
| CIFAR10 [24] | *17.5 | 17.127±0.220 | *14.73/10.739±0.016 | **10.597±0.273** | *+28.1%/**+1.3%** |
| Tiny ImageNet [25] | 47.055±3.234 | 46.221±3.655 | 31.771±3.968 | **29.492±1.296** | **+7.2%** |
| ImageNet [18] | - | - | 21.072 | **19.443** | **+7.7%** |

Our 2C loss can take advantage of both losses. Compared with the Eq. 7 loss, 2C loss considers cosine similarities of negative samples whose labels are the same as the positive image. The experimental results show that considering both *data-to-class* and *data-to-data* relations is effective and largely enhances image generation performance on CIFAR10 and Tiny ImageNet dataset. Besides, removing degenerating negative samples gives slightly better performances on CIFAR10 and Tiny ImageNet datasets.

**Comparison with other conditional GANs.** We compare ContraGAN with ACGAN [19] and ProjGAN [17], since these approaches are representative models using class information conditioning. As shown in Table 2, our approach shows favorable performances in CIFAR10, but our approach exhibits larger variations. Examples of generated images is shown in Fig. 4 (left). Experiments with Tiny ImageNet indicate that our ContraGAN is more effective when the target dataset is in the higher-dimensional space and has large inter-class variations.

**Comparison with state-of-the-art models.** We compare our method with SNResGAN [4], SAGAN [5], and BigGAN [6]. All of these approaches adopt ProjGAN [17] for class information conditioning. We conduct all experiments on Tiny ImageNet and ImageNet datasets using the hyperparameter setting used in SAGAN [5]. We use our implementation of BigGAN for a fair comparison and report the best FID values during training.

If we consider the most recent work, CRGAN [7], ICRGAN [8], and LOGAN [9] can generate more realistic images than BigGAN. Compared to such approaches, we show that our framework outperforms BigGAN by just adopting the proposed 2C loss. CRGAN and ICRGAN conduct explicit data augmentations during the training, which requires additional gradient calculations for backpropagation. Also, LOGAN needs one more feedforward and backpropagation processes for latent optimization. It takes twice as much time to train the model than standard GANs.

As a result, we identify how ContraGAN performs without data augmentations or latent optimization. Table 3 quantitatively shows that ContraGAN can synthesize images better than other state-of-the-

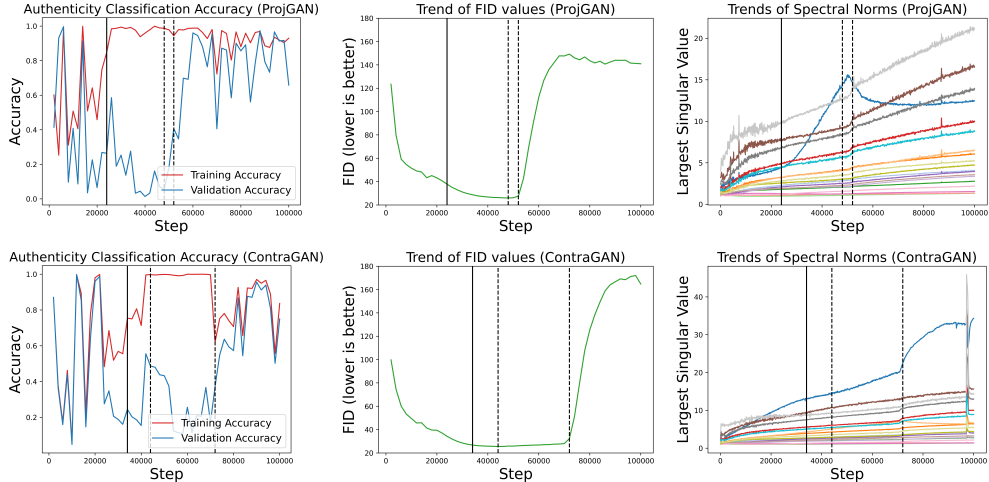

Figure 3: Authenticity classification accuracies on the training and validation datasets (left), trends of FID values (middle), and trends of the largest singular values of the discriminator's convolutional parameters (right). To specify the starting point where the difference between the training and validation accuracies is greater than 0.5, we use a solid black line. The first and second black dotted lines indicate when the performance is best and when training collapse occurs, respectively.

art GAN models under the same conditions. Compared to BigGAN, ContraGAN improves the performances by 1.3% on CIFAR10, 7.3% on Tiny ImageNet, 7.7% on ImageNet. If we use the reported number in BigGAN paper [6], the improvement is 29.9% on CIFAR10.

### 4.5 Training Stability of ContraGAN

This section compares the training stability of ContraGAN and ProjGAN [17] for the experiments using Tiny ImageNet. We compute the difference between the authenticity accuracies on the training and validation dataset. It is because the difference between training and validation performance is a good estimator for measuring the overfitting. Also, as Brock *et al.* mentioned in his work [6], the sudden change in network parameters' largest singular values (spectral norms) can indicate the collapse of adversarial training. Following this idea, we plot the trends of spectral norms of the discriminator's parameters to monitor the training collapse.

As shown in the first column of Fig. 3, ProjGAN shows the rapid increase of the accuracy difference, and ProjGAN reaches the collapse point earlier than ContraGAN. Moreover, the trend of FIDs and spectral analysis show that ContraGAN is more robust to training collapse. We speculate that ContraGAN is harder to reach undesirable status since ContraGAN jointly considers data-to-data and data-to-label relations. We discover that an increase in the accuracy on the validation dataset can indicate training collapse.

### 4.6 Ablation Study

We study how ContraGAN can be improved further with a larger batch size and data augmentations. We use ProjGAN with BigGAN architecture on Tiny ImageNet for this study. We use consistency regularization (CR) [7] to identify that our ContraGAN can benefit from regularization that uses data augmentations. Also, to identify that 2C loss is not only computationally cheap but also effective to train GANs, we replace the class embeddings with augmented positive samples (APS). APS is widely used in the self-supervised contrastive learning community [34, 49]. Table 4 shows the experiment settings, FID, and time per iteration. We indicate the number of parameters as Param. and denote three ablations – (the 2C loss, augmented positive samples (APS), and consistency regularization (CR)) as Reg.

**Large batch size.** (A, C) and (E, H) show that ContraGAN can benefit from large batch size.

**Effect of the proposed 2C loss.** (A, E) and (C, H) show that the proposed 2C loss significantly reduces FID scores of the vanilla networks (A, C) by 21.6% and 11.2%, respectively.

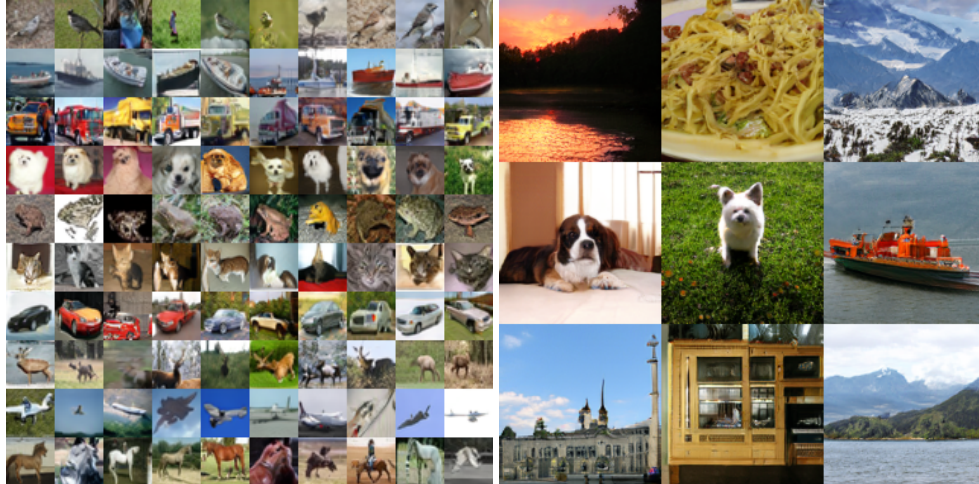

Figure 4: Examples of generated images using the proposed ContraGAN. (left) CIFAR10 [24], FID: 10.322, (right) ImageNet [18], FID: 19.443. In the case of ImageNet experiment, we select and plot well-generated images.

**Comparison with APS.** From the experiments (E, F), we can see that the 2C loss performs better than 2C loss + APS, although the latter takes about 12.9% more time. We speculate this is because each class embedding can become the representatives of the class, and it serves as the anchor that pulls corresponding images. Without the class embeddings, images in a minibatch are collected depending on a sampling state, and this may lead to training instability.

**Comparison with CR.** (A, E, G) and (C, H, I) show that vanilla + 2C loss + CR can reduce FIDs of either the results from vanilla networks (A, C) and vanilla + 2C loss (E, H). Note that the synergy is only observable if CR is used with 2C loss, and vanilla + 2C loss + CR beats vanilla + CR (B, D) with a large margin.

Table 4: Ablation study on various batch sizes, losses, and regularizations. In Reg. row, we mark − if an approach not applied and mark ✓ otherwise (in order of 2C loss, Augmented Positive Samples (APS), and Consistency Regularization [7] (CR)). Please refer Sec. 4.6 for the details.

| ID | (A) | (B) | (C) | (D) | (E) | (F) | (G) | (H) | (I) |
|---|---|---|---|---|---|---|---|---|---|
| Batch | 256 | 256 | 1024 | 1024 | 256 | 256 | 256 | 1024 | 1024 |
| Param. | 48.1 | 48.1 | 48.1 | 48.1 | 49.0 | 49.0 | 49.0 | 49.0 | 49.0 |
| Reg. | - - - | - - ✓ | - - - | - - ✓ | ✓ - - | ✓ ✓ - | ✓ - ✓ | ✓ - - | ✓ - ✓ |
| FID | 40.981 | 36.434 | 34.090 | 38.231 | 32.094 | 33.151 | 28.631 | 30.286 | **27.018** |
| Time | 0.901 | 1.093 | 3.586 | 4.448 | 0.967 | 1.110 | 1.121 | 3.807 | 4.611 |

## 5 Conclusion

In this paper, we formulate a conditional contrastive loss (2C loss) and present new Contrastive Generative Adversarial Networks (ContraGAN) for conditional image generation. Unlike previous conditioning losses, the proposed 2C loss considers not only data-to-class but also data-to-data relations between training examples. Under the same conditions, we demonstrate that ContraGAN outperforms state-of-the-art conditional GANs on Tiny ImageNet and ImageNet datasets. Also, we identify that ContraGAN helps to relieve the discriminator's overfitting problem and training collapse. As future work, we would like to theoretically and experimentally analyze how adversarial training collapses as the authenticity accuracy on the validation dataset increases. Also, we think that exploring advanced regularization techniques [8, 9, 22, 23] is necessary to understand ContraGAN further.

## Acknowledgments and Disclosure of Funding

This work was supported by Institute of Information & communications Technology Planning & Evaluation (IITP) grant funded by the Korea government (MSIT) (No.2019-0-01906, Artificial Intelligence Graduate School Program (POSTECH)). The supercomputing resources for this work was partially supported by Grand Challenging Project of Supercomuting Bigdata Center, DGIST.

## Broader Impact

We proposed a new conditional image generation model that can synthesize more realistic and diverse images. Our work can contribute to image-to-image translations [50, 51], generating realistic human faces [52, 53, 54], or any task that utilizes adversarial training.

Since conditional GANs can expand to various image processing applications and can learn the representations of high-dimensional datasets, scientists can enhance the quality of astronomical images [55, 56], design complex architectured materials [57], and efficiently search chemical space for developing materials [58]. We can do so many beneficial tasks with conditional GANs, but we should be concerned that conditional GANs can be used for deepfake techniques [59]. Modern generative models can synthesize realistic images, making it more difficult to distinguish between real and fake. This can trigger sexual harassment [60], fake news [61], and even security issues of face recognition systems [62].

To avoid improper use of conditional GANs, we need to be aware of generative models' strengths and weaknesses. Besides, it would be good to study the general characteristics of generated samples [63] and how we can distinguish fake images from unknown generative models [64, 65, 66].

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
