[Supplementary Material]

# Appendices

## A   Network Architectures

Since DCGAN [1] showed astonishing image generation ability, several generator and discriminator architectures have been proposed to stabilize and enhance the generation quality. Representatively, Miyato *et al.* [2] have used a modified version of DCGAN [1] and ResNet-style GAN [3] architectures with spectral normalization (We abbreviate it SNDCGAN and SNResGAN, respectively). Brock *et al.* [4] have expanded the capacity of SNResGAN with a shared embedding and skip connections from the noise vector (BigGAN). As a result, we tested the aforementioned frameworks to validate the proposed approach. To provide details of the main experiments in our paper, we introduce the network architectures in this section.

We start by defining some notations: $m$ is a batch size, FC(in_features, out_features) is a fully connected layer, CONV(in_channels, out_channels, kernel_size, strides) is a convolutional layer, DECONV(in_channels, out_channels, kernel_size, strides) is a deconvolutional layer, BN is a batch normalization [5], CBN is a conditional batch normalization [6, 7, 8], RELU, LRELU, and TANH indicate ReLU [9], Leaky ReLU [10], and hyperbolic tangent functions, respectively. GBLOCK(in channels, out channels, upsampling) is a generator block used in [3, 2], BIGGBLOCK(in channels, out channels, upsampling, z split dim, shared dim) is a modified version of the GBLOCK used in [4], DBLOCK(in channels, out channels, downsampling) is a discriminator block used in [4], SELF-ATTENTION is a self-attention block used in [11], NORMALIZE is a normalize operation to project given embeddings onto a unit hypersphere, and GSP is a global sum pooling layer [12]. For more details about the GBLOCK, BIGGBLOCK, DBLOCK, and the SELF-ATTENTION block, please refer to the papers [2, 11, 4] or the code of our PyTorch implementation.

Table A1: Generator of SNDCGAN [2] used for CIFAR10 [13] image synthesis.

| Layer | Input | Output | Operation |
|---|---|---|---|
| Input Layer | (m, 128) | (m, 8192) | FC(128, 8192) |
| Reshape Layer | (m, 8192) | (m, 4, 4, 512) | RESHAPE |
| Hidden Layer | (m, 4, 4, 512) | (m, 8, 8, 256) | DECONV(512, 256, 4, 2),CBN,LRELU |
| Hidden Layer | (m, 8, 8, 256) | (m, 16, 16, 128) | DECONV(256, 128, 4, 2),CBN,LRELU |
| Hidden Layer | (m, 16, 16, 128) | (m, 32, 32, 64) | DECONV(128, 64, 4, 2),CBN,LRELU |
| Hidden Layer | (m, 32, 32, 64) | (m, 32, 32, 3) | CONV(64, 3, 3, 1) |
| Output Layer | (m, 32, 32, 3) | (m, 32, 32, 3) | TANH |

Table A2: Discriminator of SNDCGAN [2] used for CIFAR10 [13] image synthesis.

| Layer | Input | Output | Operation |
|---|---|---|---|
| Input Layer | (m, 32, 32, 3) | (m, 32, 32, 64) | CONV(3, 64, 3, 1), LRELU |
| Hidden Layer | (m, 32, 32, 64) | (m, 16, 16, 64) | CONV(64, 64, 4, 2), LRELU |
| Hidden Layer | (m, 16, 16, 64) | (m, 16, 16, 128) | CONV(64, 128, 3, 1), LRELU |
| Hidden Layer | (m, 16, 16, 128) | (m, 8, 8, 128) | CONV(128, 128, 4, 2), LRELU |
| Hidden Layer | (m, 8, 8, 128) | (m, 8, 8, 256) | CONV(128, 256, 3, 1), LRELU |
| Hidden Layer | (m, 8, 8, 256) | (m, 4, 4, 256) | CONV(256, 256, 4, 2), LRELU |
| Hidden Layer | (m, 4, 4, 256) | (m, 4, 4, 512) | CONV(256, 512, 3, 1), LRELU |
| Hidden Layer | (m, 4, 4, 512) | (m, 512) | GSP |
| Output Layer | (m, 512) | (m, 1) | FC(512, 1) |

Table A3: Generator of SNResGAN [2] used for CIFAR10 [13] image synthesis.

| Layer | Input | Output | Operation |
|---|---|---|---|
| Input Layer | (m, 128) | (m, 4096) | FC(128, 4096) |
| Reshape Layer | (m, 4096) | (m, 4, 4, 256) | RESHAPE |
| Hidden Layer | (m, 4, 4, 256) | (m, 8, 8, 256) | GBLOCK(256, 256, True) |
| Hidden Layer | (m, 8, 8, 256) | (m, 16, 16, 256) | GBLOCK(256, 256, True) |
| Hidden Layer | (m, 16, 16, 256) | (m, 32, 32, 256) | GBLOCK(256, 256, True) |
| Hidden Layer | (m, 32, 32, 256) | (m, 32, 32, 3) | BN, RELU, CONV(256, 3, 3, 1) |
| Output Layer | (m, 32, 32, 3) | (m, 32, 32, 3) | TANH |

Table A4: Discriminator of SNResGAN [2] used for CIFAR10 [13] image synthesis.

| Layer | Input | Output | Operation |
|---|---|---|---|
| Input Layer | (m, 32, 32, 3) | (m, 16, 16, 128) | DBLOCK(3, 128, True) |
| Hidden Layer | (m, 16, 16, 128) | (m, 8, 8, 128) | DBLOCK(128, 128, True) |
| Hidden Layer | (m, 8, 8, 128) | (m, 8, 8, 128) | DBLOCK(128, 128, False) |
| Hidden Layer | (m, 8, 8, 128) | (m, 8, 8, 128) | DBLOCK(128, 128, False), RELU |
| Hidden Layer | (m, 8, 8, 128) | (m, 128) | GSP |
| Output Layer | (m, 128) | (m, 1) | FC(128, 1) |

Table A5: Generator of BigGAN [4] used for CIFAR10 [13] image synthesis.

| Layer | Input | Output | Operation |
|---|---|---|---|
| Input Layer | (m, 20) | (m, 6144) | FC(20, 6144) |
| Reshape Layer | (m, 6144) | (m, 4, 4, 384) | RESHAPE |
| Hidden Layer | (m, 4, 4, 384) | (m, 8, 8, 384) | BIGGBLOCK(384, 384, True, 20, 128) |
| Hidden Layer | (m, 8, 8, 384) | (m, 16, 16, 384) | BIGGBLOCK(384, 384, True, 20, 128) |
| Hidden Layer | (m, 16, 16, 384) | (m, 16, 16, 384) | SELF-ATTENTION |
| Hidden Layer | (m, 16, 16, 384) | (m, 32, 32, 384) | BIGGBLOCK(384, 384, True, 20, 128) |
| Hidden Layer | (m, 32, 32, 384) | (m, 32, 32, 3) | BN, RELU, CONV(384, 3, 3, 1) |
| Output Layer | (m, 32, 32, 3) | (m, 32, 32, 3) | TANH |

Table A6: Discriminator of BigGAN [4] used for CIFAR10 [13] image synthesis.

| Layer | Input | Output | Operation |
|---|---|---|---|
| Input Layer | (m, 32, 32, 3) | (m, 16, 16, 192) | DBLOCK(3, 192, True) |
| Hidden Layer | (m, 16, 16, 192) | (m, 16, 16, 192) | SELF-ATTENTION |
| Hidden Layer | (m, 16, 16, 192) | (m, 8, 8, 192) | DBLOCK(192, 192, True) |
| Hidden Layer | (m, 8, 8, 192) | (m, 8, 8, 192) | DBLOCK(192, 192, False) |
| Hidden Layer | (m, 8, 8, 192) | (m, 8, 8, 192) | DBLOCK(192, 192, False) |
| Hidden Layer | (m, 8, 8, 192) | (m, 192) | RELU, GSP |
| Output Layer | (m, 192) | (m, 1) | FC(192, 1) |

Table A7: Generator of BigGAN [4] for Tiny ImageNet [14] image synthesis.

| Layer | Input | Output | Operation |
|---|---|---|---|
| Input Layer | (m,20) | (m,20480) | FC(20, 20480) |
| Reshape Layer | (m,20480) | (m,4,4,1280) | RESHAPE |
| Hidden Layer | (m,4, 4, 1280) | (m,8, 8, 640) | BIGGBLOCK(1280, 640, True, 20, 128) |
| Hidden Layer | (m,8, 8, 640) | (m,16, 16, 320) | BIGGBLOCK(640, 320, True, 20, 128) |
| Hidden Layer | (m,16, 16, 320) | (m,32, 32, 160) | BIGGBLOCK(320, 160, True, 20, 128) |
| Hidden Layer | (m,32, 32, 160) | (m,32, 32, 160) | SELF-ATTENTION |
| Hidden Layer | (m,32, 32, 160) | (m,64, 64, 80) | BIGGBLOCK(160, 80, True, 20, 128) |
| Hidden Layer | (m,64, 64, 80) | (m,64, 64, 3) | BN, RELU, CONV(80,3, 3, 1) |
| Output Layer | (m,32, 32, 3) | (m,32, 32, 3) | TANH |

Table A8: Discriminator of BigGAN [4] for Tiny ImageNet [14] image synthesis.

| Layer | Input | Output | Operation |
|---|---|---|---|
| Input Layer | (m, 64, 64, 3) | (m, 32, 32, 80) | DBLOCK(3, 80, True) |
| Hidden Layer | (m, 32, 32, 80) | (m, 32, 32, 80) | SELF-ATTENTION |
| Hidden Layer | (m, 32, 32, 80) | (m, 16, 16, 160) | DBLOCK(80, 160, True) |
| Hidden Layer | (m, 16, 16, 160) | (m, 8, 8, 320) | DBLOCK(160, 320, True) |
| Hidden Layer | (m, 8, 8, 320) | (m, 4, 4, 640) | DBLOCK(320, 640, True) |
| Hidden Layer | (m, 4, 4, 640) | (m, 4, 4, 1280) | DBLOCK(640, 1280, False) |
| Hidden Layer | (m, 4, 4, 1280) | (m, 1280) | RELU, GSP |
| Output Layer | (m, 1280) | (m, 1) | FC(1280, 1) |

Table A9: Generator of BigGAN [4] for ImageNet [15] image synthesis.

| Layer | Input | Output | Operation |
|---|---|---|---|
| Input Layer | (m,20) | (m,24576) | FC(20, 24576) |
| Reshape Layer | (m,24576) | (m,4,4,1536) | RESHAPE |
| Hidden Layer | (m,4,4,1536) | (m,8,8,1536) | BIGGBLOCK(1536, 1536, True, 20, 128) |
| Hidden Layer | (m,8,8,1536) | (m,16,16,768) | BIGGBLOCK(1536, 768, True, 20, 128) |
| Hidden Layer | (m,16,16,768) | (m,32,32,384) | BIGGBLOCK(768, 384, True, 20, 128) |
| Hidden Layer | (m,32,32,384) | (m,64,64,192) | BIGGBLOCK(384, 192, True, 20, 128) |
| Hidden Layer | (m,64,64,192) | (m,64,64,192) | SELF-ATTENTION |
| Hidden Layer | (m,64,64,192) | (m,128,128,96) | BIGGBLOCK(192, 96, True, 20, 128) |
| Hidden Layer | (m,128,128,96) | (m,128,128,3) | BN, RELU, CONV(96, 3, 3, 1) |
| Output Layer | (m,128,128,3) | (m,128,128,3) | TANH |

Table A10: Discriminator of BigGAN [4] for ImageNet [15] image synthesis.

| Layer | Input | Output | Operation |
|---|---|---|---|
| Input Layer | (m, 128, 128, 3) | (m, 64, 64, 96) | DBLOCK(3, 96, True) |
| Hidden Layer | (m, 64, 64, 96) | (m, 64, 64, 96) | SELF-ATTENTION |
| Hidden Layer | (m, 64, 64, 96) | (m, 32, 32, 192) | DBLOCK(96, 192, True) |
| Hidden Layer | (m, 32, 32, 192) | (m, 16, 16, 384) | DBLOCK(192, 384, True) |
| Hidden Layer | (m, 16, 16, 384) | (m, 8, 8, 768) | DBLOCK(384, 768, True) |
| Hidden Layer | (m, 8, 8, 768) | (m, 4, 4, 1536) | DBLOCK(768, 1536, True) |
| Hidden Layer | (m, 4, 4, 1536) | (m, 4, 4, 1536) | DBLOCK(1536, 1536, False) |
| Hidden Layer | (m, 4, 4, 1536) | (m, 1536) | RELU, GSP |
| Output Layer | (m, 1536) | (m, 1) | FC(1536, 1) |

# B Hyperparameter Setup

Table A11: Hyperparameter values used for experiments. Settings (B, C, E) and (F) are the settings used in [16, 1, 17] and [11], respectively. we conduct experiments with CIFAR10 [13] using the settings (A, B, C, D, E) and with Tiny ImageNet [14] and ImageNet [15] using the setting (F).

| Setting | $\alpha_1$ | $\alpha_2$ | $\beta_1$ | $\beta_2$ | $n_{dis}$ |
|---|---|---|---|---|---|
| A | 0.0001 | 0.0001 | 0.5 | 0.999 | 2 |
| B | 0.0001 | 0.0001 | 0.5 | 0.999 | 1 |
| C | 0.0002 | 0.0002 | 0.5 | 0.999 | 1 |
| D | 0.0002 | 0.0002 | 0.5 | 0.999 | 2 |
| E | 0.0002 | 0.0002 | 0.5 | 0.999 | 5 |
| F | 0.0004 | 0.0001 | 0.0 | 0.999 | 1 |

Choosing a proper hyperparameter setup is crucial to train GANs. In this paper, we conduct experiments using six settings with Adam optimizer [18]. $\alpha_1$ and $\alpha_2$ are the learning rates of the discriminator and generator. $\beta_1$ and $\beta_2$ are the hyperparameters of Adam optimizer to control exponential decay rates of moving averages. $n_{dis}$ is the number of discriminator iterations per single generator iteration. For the contrastive coefficient $\lambda$ (see Algorithm 1), the value is fixed at 1.0 for a fair comparison with [19, 8]. In all experiments, we use the temperature $t = 1.0$. Experiments over temperature are displayed in Fig. A1. Besides, we apply moving average update of the generator's weights used in [20, 21, 22] after 20,000 generator iterations with the decay rate of 0.9999. The settings (B, C, E) are known to give satisfactory performances on CIFAR10 [13] in previous papers [16, 1, 17]. Since Heusel *et al.* [23] and Zhang *et al.* [11] have shown that two

Figure A1: Change of FID values as the temperature increases. Experiments are executed three times, and the means and standard deviations are represented by the blue dots and solid lines, respectively.

time-scale update (TTUR) can converge to a stationary local Nash equilibrium [24], we adopt the hyperparameter setup used in [11] (setting F) to generate realistic images on Tiny ImageNet [14] and ImageNet [15] datasets.

**Experimental setup used for Table 1 in the main paper**: Experiments on CIFAR10 dataset are conducted three times with different random seeds using the setting (E) with the batch size of 64 until 80k generator updates. Experiments on Tiny ImageNet dataset are performed three times until 100k generator updates using the setting (F) with the batch size of 256 and BigGAN architecture (see Table A7 and Table A8).

**Experimental setup used for Table 2 in the main paper**: Experiments on CIFAR10 dataset are performed three times with different random seeds using the settings (A, B, C, D, E) with the batch size of 64. We stop training GANs with SNDCGAN, SNResGAN, and BigGAN architectures after 200k, 100k, and 80k generator updates, respectively. Also, we report performances of the hyperparameter settings that showed the lowest FID values by mean. Experiments on Tiny ImageNet dataset are conducted three times until 100k generator updates using the setting (F) with the batch size of 256 and BigGAN architecture (see Table A7 and Table A8). The hyperparameter settings: C, D, E, show the best performance in SNDCGAN [2], SNResGAN [2], and BigGAN [4], respectively. We reason that as the model's capacity increases, training GANs becomes more difficult; thus, it requires more discriminator updates. Moreover, we experimentally identify that updating discriminator more times does not always produce better performance, but it might be related to the model capacity.

**Experimental setup used for Table 3 in the main paper**: FID values on CIFAR10 dataset are reported using the setting (E) with the batch size of 64. The experiments on the Tiny ImageNet are conducted using the setting (F) with the batch size of 1024. Experiments on ImageNet dataset are executed once until 250k generator updates using the setting (F) with the batch size of 256 and BigGAN architecture (see Table A9 and Table A10). All other settings not noticed here are the same as the experimental setup for Table 2 above.

**Experimental setup used for Table 4 in the main paper**: All ablation results are reported using the setting (F), and models with consistency regularization (CR) [17] are trained with the coefficient of 10.0. We use an Intel(R) Xeon(R) Silver 4114 CPU, four NVIDIA Geforce RTX 2080 Ti GPUs, and PyTorch DataParallel library to measure time per iteration. All other settings not noticed here are the same as the experimental settings used for Table 2.

## C   Nonlinear Projection and Batch Size

We study the effect of a projection layer $h : \mathbb{R}^k \longrightarrow \mathbb{S}^d$ that is introduced in Sec. 3.2. We change the types of the layer (linear vs. nonlinear) and increase the dimensionality of projected embeddings, $d$ on CIFAR10 dataset. Fig. A2a shows the overview of FID values. All experiments are conducted using three different architectures: DCGAN, ResGAN, and BigGAN that are equipped with spectral normalization. We also run the experiments using three different random seeds and do not apply moving average update of the generator's weights. SNDCGAN with the liner projection layer projects

|     |     |
|:---:|:---:|
| (a) | (b) |

Figure A2: (a) FID values of ContraGANs with different projection layers and embedding dimensionalities. (b) The change in FID values as the batch size increases. The experiments (a) and (b) are conducted using the setting (D).

latent features onto the 1,024 dimensional space. This configuration shows higher FID than the nonlinear counterpart, but ContraGANs with a linear projection layer generally give lower FIDs. Although GANs are known to need careful hyperparameter selection, our ContraGAN does not seem to be sensitive to the type and dimensionality of the projection layer.

Figure A2b shows the change in FID values as the batch size increases. Experiments conducted by Brock *et al.* [4] have demonstrated that increasing the batch size enhances image generation performance on ImageNet dataset [15]. However, as shown in Fig. A2b, optimal batch sizes for CIFAR10 and Tiny ImageNet are 64 and 1,024, respectively. Based on these results, we can deduce that increasing batch size does not always give the best synthesis results. We presume that this phenomenon is related to the number of classes used for the training.

# D    FID Implementations

Table A12: Comparison of TensorFlow and PyTorch FID implementations.

|     | ContraGAN | |
|-----|-----------|-----------------|
| FID implementation | CIFAR10 [13] | Tiny ImageNet [14] |
| TensorFlow [25] | 10.308 | 26.924 |
| PyTorch [26] | 10.304 | 27.131 |

FID is a widely used metric to evaluate the performance of a GAN model. Since calculating FID requires a pre-trained inception-V3 network [27], many implementations use Tensorflow [28] or PyTorch [29] libraries. Among them, the TensorFlow implementation [25] for FID measurement is widely used. We use the PyTorch implementation for FID measurement [26], instead. In this section, we show that the PyTorch-based FID implementation [26] used in our work provides almost the same results as the TensorFlow implementation. The results are summarized in Table A12.

# E    Multiple Runs of the Stability Experiment

In this section, we provide the additional results of the stability test performed in Sec. 4.5 of the main paper. The third and fourth row of Fig. A3 shows the another run from ProjGAN and ContraGAN.

Figure A3: Authenticity classification accuracies on the training and validation datasets (left), trends of FID values (middle), and trends of the largest singular values of the discriminator's convolutional parameters (right). To specify the starting point where the difference between the training and validation accuracies is greater than 0.5, we use a solid black line. The first and second black dotted lines indicate when the performance is best and when training collapse occurs, respectively.

As shown in the third row of Fig. A3, training collapse does not occur in training ProjGAN [8]. However, the best FID value of the ProjGAN is 34.831, which is much higher than that of ContraGAN ($25 \leq$ FID $\leq 27$). The above results show that ContraGAN is more robust to the overfitting and training collapse.

# F   Qualitative Results

This section presents images generated by various conditional image generation frameworks. Fig. A4, A5, and A6 show the synthesized images using CIFAR10 dataset. Fig. A7 and A8 show the synthesized images using Tiny ImageNet dataset. Fig. A9 and A10 show the generated images using ImageNet dataset. As shown in Fig. A8 and A10, our approach can achieve favorable FID compared to the other baseline approaches.

Figure A4: Examples generated by ACGAN [19] trained on CIFAR10 dataset [13] (FID=11.111).

Figure A5: Examples generated by ProjGAN [8] on CIFAR10 dataset [13] (FID=10.933).

Figure A6: Examples generated by ContraGAN (Ours) on CIFAR10 dataset [13] (FID=10.188).

Figure A7: Examples generated by ProjGAN [8] on Tiny ImageNet dataset [14] (FID=34.090).

Figure A8: Examples generated by ContraGAN (Ours) on Tiny ImageNet dataset [14] (FID=30.286).

Figure A9: Examples generated by ProjGAN [8] on ImageNet dataset [15] (FID=21.072).

Figure A10: Examples generated by ContraGAN (Ours) on ImageNet dataset [15] (FID=19.443).