[Reviews · NeurIPS 2020]

Review 1

Summary and Contributions: The authors propose a class conditional contrastive loss for conditional image synthesis by GANs (ContraGAN). Specifically, the proposed class conditional contrastive loss is designed to maximize the mutual information of images from the same class. The authors show that ContraGAN outperforms state-of-the-art GANs by significant improvements in terms of FID in CIFAR10 and Tiny ImageNet.

Strengths: 1. The idea of designing contrastive learning for conditional image synthesis is interesting and the proposed formulation is somehow novel. 2. Experiments show strong improvements in terms of FID in CIFAR10 and Tiny ImageNet.

Weaknesses: 1. The motivation is not clear. As claimed in Line 34-36, ContraGAN seems to target at solving the unstable training problem in conditional image synthesis. However, the paper does not explain why the proposed model can learn data-to-data relations in ContraGAN to stabilize GAN’s training. Besides, the paper does not provide stability comparisons to verify the improvements in terms of stabilizing training. Specifically, the authors can visualize the singular value of models following BigGAN [6] for comparisons. 2. Important experiments are missing. Since the XT-Xent [25] (Eq. (6)) is also able to learn data-to-data relations, the authors need to compare experimental results from different types of contrastive losses, including Eq. (6), Eq. (7), and Eq. (8), to verify the effectiveness of the proposed 2C loss (Eq. (8)). 3. The experimental setting can be biased. All the baselines, including SNResGAN [4], SAGAN [5], BigGAN [6], ACGAN [19], cGAN [17], conduct experiments on ImageNet, but this paper uses Tiny ImageNet (which is not acceptable). Besides, this paper uses 64x64 for training and testing, the resolution could be too small compared with 128x128 resolution used in SNResGAN, SAGAN, BigGAN, ACGAN and cGAN.

Correctness: The claims and the method are correct. The empirical methodology is correct.

Clarity: The paper can be improved further. Some important details are missing. 1. What is the data-to-data relation? The authors should provide a detailed explanation or cite [38] in the Introduction section when it is first mentioned. 2. Fig. 1 is confusing. A detailed introduction of (a-e) is missing. The authors should also provide detailed comparisons between (f) and (a-e). 3. It would be better to introduce some related works of metric learning in GAN (e.g., InfoGAN, Chen et al., NeurIPS 2016) instead of the vanilla GAN [1] in Section 2.

Relation to Prior Work: The differences between this work and some previous contributions are discussed in this paper. However, some discussions are not clearly enough (more details refer to the previous question).

Reproducibility: Yes

Additional Feedback:


Review 2

Summary and Contributions: The paper proposes ContraGAN, a conditional GAN framework that leverages contrastive learning to improve image quality compared to other conditional GAN variants. The proposed approach maximizes the lower-bound on mutual information using a novel conditional contrastive loss (2C loss), which considers both data-to-class and data-to-data relationships.

Strengths: - The proposed method is theoretically sound and well motivated. The idea of using a 2C loss is novel and could be useful for the community. - The 2C loss can be successfully combined with other regularization techniques, such as CR as shown in Table 3. - The ContraGAN model shows improvement in class conditional image synthesis on Tiny ImageNet in comparison to other methods.

Weaknesses: - In Table 1, for Cifar 10, the differences between cGAN and ContraGAN are smaller than the standard deviations. Therefore, the claim that ContraGAN outperforms the cGAN with projection discriminator is not fully supproted. Only the experiment on TinyImageNet reveals an improvement in performance. The same holds true for Table 2: The difference in FID to BigGAN is 0.4, and this does not yet take the standard deviation into account, which was omitted in this case. Since improved performance is only shown for a single run (line 297) on a single dataset (Tiny ImageNet) the empirical evidence for the claim that the contrastive loss works better than the projection loss is weak. - The paper states that GAN models were reimplemented (line 199). In Table 2 the value for SNResGAN on Cifar10 is taken from the original paper, however for the same setup there is a number for the re-implementation in Table 1 and seems to be better. This is confusing. - The experimental evaluation is only for low-resolution image synthesis task (32x32 and 64x64 images). It would be beneficial to see the performance of the proposed approach on higher resolution images. - The discussion on the training stability of the proposed approach is missing. It is well known that GANs suffer from training instability, e.g. for BigGAN roughly half of the training runs is unsuccessful, leading to mode collapses and other issues. How the use of 2C loss influences the training stability?

Correctness: The claim that ContraGAN outperforms other models on Cifar10 is questionable (line 13, 276). The paper claims that the 9 reimplementations perform better than the original (line 65), however, that appears to just hold true for 2 methods (line 202).

Clarity: The paper communicates the idea clearly, both in written form and via the figures. Nevertheless, the writing can be improved. E.g. the paper refers to “cGAN with a projection discriminator” as just “cGAN” (line 88, 227). This is confusing, since “cGAN” denotes a wide class of conditional GANs, starting with [1]. [1] Mirza, Mehdi, and Simon Osindero. "Conditional generative adversarial nets." arXiv preprint arXiv:1411.1784 (2014).

Relation to Prior Work: The difference to prior related work (projection discriminator, ACGAN) is clearly described.

Reproducibility: Yes

Additional Feedback: - Line 215: It is not clear how concatenating noise to y makes the comparison unfair. ==== Post-rebuttal feedback ==== I have carefully read the reviews and the response the authors provided to the raised concerns and comments. I appreciate that the authors have responded and addressed several remarks/questions I had. Some inconsistencies in the reported numbers in Table 1 and 2 have been clarified for me. Updated numbers in Table 1 and 2 in the author's response show only minor improvement over BigGAN, however, the 2C loss shows superior performance in comparison to projection on Tiny ImageNet and the results on ImageNet at 128x128 look promising. I'm still missing the discussion on the training stability, as in their response the authors only analyze one run, where both BigGAN and their model collapse at some point in time, though ContraGAN at a later stage. It would be good to look at multiple runs to analyze stability of the two models, in particular as both models show very similar performance on CIFAR10 and Tiny ImageNet. Overall, I'm willing to raise my score to 6 if the provided clarifications, updated numbers and adjusted claims, as well as final results on ImageNet are added to the paper.


Review 3

Summary and Contributions: This paper proposed a new conditional GAN, Contrastive Generative Adversarial Networks(ContraGAN), using a new conditional contrastive loss. It formulated a lower bound on mutual information between image features and proposed a novel conditional contrastive loss (2C loss) based on this. Experiments showed the proposed method generates more realistic and diverse results than STOA.

Strengths: 1. The paper proposed a new conditional contrastive loss, to pull embeddings of same class images and push away embeddings of different class images. In comparison, previous ACGAN an cGAN only pull same class embedding. 2. The proposed contrastive loss is based on an understanding of lower bounds on mutual information between images. And a proposition on lower bounds of mutual information is proved. 3. Extensive experiments (including quantitative evaluation, qualitative evaluation and ablation studies) validate the proposed method's effectiveness. 4. The software developed for fair comparison of state-of-the-art approaches is useful to the community and can facilitate follow-up research.

Weaknesses: 1. The new conditional contrastive loss is largely inspired by XT-Xent loss, and the formula are quite close (i.e. replacing data augmentation with embeddings of class labels and adding cosine similarities of negative samples). This makes the novelty limited. 2. The evaluation only uses one GAN metric (FID). To validate the diversity of generated results, other GAN metrics related to generation diveristy could be adopted.

Correctness: The proposed proposition1 on lower bounds on mutual information of image features has been proved theoretically. And the new contrastive loss is developed based on the similarity between proposition1 and metric learning. So I think the claims and method are theoretically sound and correct.

Clarity: The paper writing is clear and easy to follow. The illustrations also help clarify the differences between the proposed method and some previous works.

Relation to Prior Work: The difference between ACGAN, cGAN and ContraGAN has been clearly discussed. But I think the difference between XT-Xent and ContraGAN has not been clearly discussed.

Reproducibility: Yes

Additional Feedback: It would be better to provide more details about the difference between XT-Xent and the proposed loss. ============== Post-rebuttal feedback =================== Rebuttal addressed some of my concerns. Rebuttal explained more about 2C loss and difference between 2C loss and XT-Xent, but it still not vey clear to me. I suggest to make the above concepts more clear. But overall I think the paper is above the acceptance threshold and I keep my original rating.


Review 4

Summary and Contributions: The paper presents new way for conditioning discriminator on class labels based on contrastive loss.

Strengths: - The paper contain strong evaluation part, authors spend a lot of afford to make a fair comparisons with sota methods. - Authors evaluate different architectures and show that their method provide a consistent improvement. - The paper contain thoughtful ablation, which include recent consistency regularization scheme.

Weaknesses: - The main motivation of the paper is the specific organization of the latent space. E.g. images with the same class label should have similar embedding in the discriminator space. This motivations is valid for regular classification problems, where we need to separate different classes. While in case of gans we have real and fake images, and the task of discriminator is to separate these two given the class label. So in other words it is not clear how the separation of the classes should help discriminating real and fake images. - Moreover the proposed loss seems to suffer from the same problem as ACGAN, e.g. tendency to generate images that is easily classifiable. Indeed L2C loss is minimized when all images from the same class have the same embedding. So the generator will tend to produce images that has embedding similar to e(y_i). - Some parts of the paper is not clear (see clarity).

Correctness: The main motivation seems week (see weaknesses), while the empirical evaluation looks correct (see strength).

Clarity: 1) Fig. 1 is completely non intuitive and overwhelmed with different unintuitive symbols. - For example in XT-Xent[25] only augmented images are used (according to notation diamonds), while in Fig 1(c) there a lot of squares. Also it is not clear why one of the gray diamond is pushed towards circle, while other pushed away from it. - Also the Fig.1 is placed in the introduction, while (a), (b) and (c) parts is not introduced anywhere in the introduction. This makes pretty hard for the reader to understand what is these methods, and why they are relevant. - Moreover cGAN (e) seem incorrect. Reference (circle) did not pulled towards hypothetical class label embedding. Class embedding here act as separation hyperplane which separate fake images (of particular class) from the real ones. 2) The connection between Eq. (7) and Eq. (8) is not clear. Why the Eq. (7) should push negative having the same label y_i? The optimum for this objective is reached when all class e(y_i) is orthogonal and l(x_i) is equal to e(y_i). Eq. (8) has the same optimum. 3) The connection between mutual information, metric learning and Eq. (8) is a bit messy. What is the purpose of introducing mutual information and then immediately replacing it with a metric learning intuition? Also there is no theoretical connection between mutual information and Eq. (8), just a metric learning intuition. So one could conclude that the mutual information part is misleading and it is not useful for understanding. 4) There are some minor typos: Line 85: cGAN points out -> Autors of cGAN[17] point out Line 279: conditional batch normalization -> conditional coloring transform

Relation to Prior Work: The contribution of the work is the new loss for conditional GAN training. The loss seems novel in the context of conditional GANs.

Reproducibility: Yes

Additional Feedback: While I understand that complicated benchmarks such as ImageNet may be complicated to handle, I suggest the authors to make an effort to include it in the future version of this work. Otherwise the paper risk to be unnoticed by the community. -------------------------------------------------------------------------------------------------------------- Rebuttal did not change my opinion. It is definitely good that image-net experiments are added. However from the rebuttal I did not understand how the motivation will be explained and response to '(R4) May generate images that are easily classifiable.' make no sense to me. I suggest to work more on intuition and motivation part of the paper.

[Author Response · NeurIPS 2020]

We thank the reviewers for the constructive comments. Reviewers appreciate the effectiveness of the proposed
ContraGAN (R1, R2, R3, R4), the novelty of the proposed 2C loss (R1, R2, R4), composability with modern
regularization techniques (R2, R4), and usefulness of our software (R3). This rebuttal answers questions raised by
reviewers. Every experiment and explanation in this rebuttal will be included in the paper.

**(R1, R2, R4) Clarity.** We will introduce the concept of data-to-data relations carefully. Symbols in Fig 1 will be
polished, as suggested. cGAN with a projection discriminator [17] will be named as *ProjGAN* to avoid confusion. We
will make smooth transitions among mutual information, metric learning, contrastive loss, and 2C loss.

**(R1) Comparison with other metric learning losses shown in Fig. 1.** P-NCA loss [24] does not explicitly look at
data-to-data relations, and XT-Xent loss [25] (equivalent to Eq. 6) does not take account of data-to-label relations.
Our 2C loss can take advantage of the strengths of both losses. Compared with Eq. 7 loss, 2C loss considers cosine
similarities of negative samples. We conduct experiments to compare 2C loss with other losses. Every experiment is
   performed three times, and its mean±variance of FID [39] is reported below.

| Dataset | Unconditional GAN [6] | with P-NCA loss [24] | with Eq.6 loss (XT-Xent) [25] | with Eq.7 loss | ContraGAN |
|---|---|---|---|---|---|
| CIFAR10 [21] | 15.550±1.955 | 15.350±0.017 | 14.832±0.695 | 10.886±0.072 | **10.597±0.273** |
| Tiny ImageNet [22] | 56.297±1.499 | 47.867±1.813 | 54.394±9.982 | 33.488±1.006 | **32.720±1.084** |

**(R3) Difference between XT-Xent loss and 2C loss.** XT-Xent is intended for unsupervised learning, and XT-Xent
only regards the augmented images as the positive samples. On the other hand, 2C loss utilizes weak supervision from
label information. Therefore, compared with 2C loss, XT-Xent hardly gathers image embeddings of the same class. The
table above shows the effectiveness of 2C loss. Besides, XT-Xent loss requires extra data augmentations and additional
forward/backward propagations, which induce $15 \sim 20\%$ more training time than using 2C loss.

**(R1, R2) Reliability of experiments.** We provide updated Table 1 (left) and 2 (right) below after three times of
experiments. To avoid the single-trial analysis, we will replace the original tables with these numbers. As pointed out
   by R2, we will mention that ProjGAN is on par with ContraGAN in CIFAR10 dataset.

| Dataset/Batch Size/Res. | ACGAN [19] | ProjGAN [17] | ContraGAN | Dataset/Batch Size/Res. | SNResGAN [4] | SAGAN [5] | BigGAN [6] | ContraGAN |
|---|---|---|---|---|---|---|---|---|
| CIFAR10/64/32 | 10.697±0.129 | 10.739±0.016 | **10.597±0.273** | CIFAR10/64/32 | *17.5 | 17.127±0.220 | 10.739±0.016 | **10.597±0.273** |
| Tiny ImageNet/256/64 | 88.628±5.523 | 37.563±4.390 | **32.720±1.084** | Tiny ImageNet/1024/64 | 47.055±3.234 | 46.221±3.655 | 31.771±3.968 | **29.492±1.296** |

**(R1, R2, R4) ImageNet.** We perform ImageNet [18] experiments. It has not been completed within six days of the
rebuttal period, and it reaches 160k iterations. We compare SAGAN and ProjGAN here, since we were able to get FID
of those approaches for 160k iterations. Under the same iteration number, FID and synthesized images by ContraGAN
   is quite promising. Note that 160k iterations are pretty early stage, since 10M iterations are often applied.

| Experiment with ImageNet with batch size 256. Image Res. $128 \times 128$, iteration 160k / 10M. | | |  |
|---|---|---|---|
| SAGAN | ProjGAN | ContraGAN | |
| 40.96 [5] | 32.456 | **26.935** | |

**(R2) Inconsistent SNResGAN results in Table 1 and 2.** We found that SNResGAN [4] and SAGAN [5] can be
improved by applying the moving average update (MAU) for the generator's weights (described in Sec. 3 in the
supplement). We use our implementation for every result in Table 1, and we apply the MAU to report the best results.
Table 2 takes the numbers from the original paper [4], so we did not use the MAU. This makes inconsistency.

**(R3) Diversity of generated images.** Since FID [39] can measure both fidelity and diversity of images, we claim that
ContraGAN can generate more diverse images compared with the previous methods. For more analysis, Intra-FID [17]
can be adopted to measure the degree of intra-class variation.

**(R4) May generate images that are easily classifiable.** ContraGAN can look at the condition of inside the batch and
decide the authenticity of images using relations of examples. Thus, if the generator gives images from a restricted
mode to the discriminator, the discriminator can recognize the generated samples as the fake using the relations. This
procedure can lead the generator to create more diverse images to deceive the discriminator.

**(R1, R2) Stability.** The right figures show the singu-
lar values of convolutional layers. We observe mode-
collapse of ProjGAN at 45k steps, whereas ContraGAN
runs 72k steps without mode-collapse. We speculate that
ContraGAN is harder to reach undesirable status, since
ContraGAN jointly considers data-to-data and data-to-
label relations. Tiny ImageNet dataset is used for this
experiment.



[Meta-Review · NeurIPS 2020]

Reviewers were split on this paper with three recommending accept and one recommending reject. The main concerns were missing experiments on ImageNet and lack of clarify on why the method should work, particularly with regard to how it stabilizes training. After the rebuttal, the reviewers and AC were more confident in the experimental results and recommend acceptance, but the authors are urged to 1) complete the full experiments on ImageNet, 2) analyze stability over multiple runs and provide some discussion of why the proposed method should help stability. Also please see the other detailed recommendations in the reviews.